# Antimicrobial Peptides Designed against the Ω-Loop of Class A β-Lactamases to Potentiate the Efficacy of β-Lactam Antibiotics

**DOI:** 10.3390/antibiotics12030553

**Published:** 2023-03-10

**Authors:** Sarmistha Biswal, Karina Caetano, Diamond Jain, Anusha Sarrila, Tulika Munshi, Rachael Dickman, Alethea B. Tabor, Surya Narayan Rath, Sanjib Bhakta, Anindya S. Ghosh

**Affiliations:** 1Molecular Microbiology Laboratory, Department of Biotechnology, Indian Institute of Technology Kharagpur, Kharagpur 721302, West Bengal, India; 2Mycobacteria Research Laboratory, Institute of Structural and Molecular Biology, Department of Biological Sciences, Birkbeck, University of London, Malet Street, London WC1E 7HX, UK; 3Department of Chemistry, University College London, Gordon Street, London WC1H 0AJ, UK; 4School of Pharmacy, University College London, Brunswick Square, London WC1N 1AX, UK; 5Department of Bioinformatics, Odisha University of Agriculture and Technology, Bhubaneswar 751003, Odisha, India

**Keywords:** β-lactamases, antimicrobial peptide (AMP), Ω-loop, peptide modelling, docking, MD simulation, solid phase synthesis, HR-mass spectrometry, HT-SPOTi, enzyme kinetics

## Abstract

Class A serine β-lactamases (SBLs) have a conserved non-active site structural domain called the omega loop (Ω-loop), in which a glutamic acid residue is believed to be directly involved in the hydrolysis of β-lactam antibiotics by providing a water molecule during catalysis. We aimed to design and characterise potential pentapeptides to mask the function of the Ω-loop of β-lactamases and reduce their efficacy, along with potentiating the β-lactam antibiotics and eventually decreasing β-lactam resistance. Considering the Ω-loop sequence as a template, a group of pentapeptide models were designed, validated through docking, and synthesised using solid-phase peptide synthesis (SPPS). To check whether the β-lactamases (BLAs) were inhibited, we expressed specific BLAs (TEM-1 and SHV-14) and evaluated the trans-expression through a broth dilution method and an agar dilution method (HT-SPOTi). To further support our claim, we conducted a kinetic analysis of BLAs with the peptides and employed molecular dynamics (MD) simulations of peptides. The individual presence of six histidine-based peptides (TSHLH, ETHIH, ESRLH, ESHIH, ESRIH, and TYHLH) reduced β-lactam resistance in the strains harbouring BLAs. Subsequently, we found that the combinational effect of these peptides and β-lactams sensitised the bacteria towards the β-lactam drugs. We hypothesize that the antimicrobial peptides obtained might be considered among the novel inhibitors that can be used specifically against the Ω-loop of the β-lactamases.

## 1. Introduction

Multi-drug-resistant strains of a selection of WHO-priority infectious bacteria are creating an alarming situation and have claimed millions of lives globally [1]. This is anticipated to result in 10 million fatalities annually by 2050 if no new antimicrobial approaches are implemented [2]. The production of β-lactamases is the primary cause of β-lactam resistance [3]. β-lactamases are enzymes that are evolutionarily related to penicillin-binding proteins (PBPs) and protect bacterial cells by hydrolysing the β-lactam ring of the β-lactam antibiotics, thus making them ineffective. The β-lactamases are classified into four different groups based on their amino acid sequences, i.e., classes A, C, and D contain evolutionarily distinct serine β-lactamases, and class B metallo-enzymes contain zinc ions. All the serine beta-lactamases possess a distinct loop structure called the omega loop which is a non-regular structural motif that resembles the upper case Greek letter omega (Ω), and it is essential for stabilizing enzyme–substrate interaction. Specifically, the conserved ῼ-loop of class A β-lactamases has been proven to play a crucial role in the hydrolytic reaction during β-lactam hydrolysis [4]. The glutamic acid of the ῼ-loop (Figure 1) is known to facilitate the movement of the water molecule required for degrading the β-lactams [5,6]. However, improper use or overuse of β-lactams has resulted in the failure of standard clinical treatments [7]. On the other hand, antimicrobial peptides (AMPs) can be suitable leads for future drugs as they are promising alternatives to conventional antibiotics with high selectivity and efficacy [8,9]. Due to low chemical and physical stability and a short plasma half-life, more than 7000 naturally occurring peptides have limited usage [10]. The US Food and Drug Administration (FDA) has approved more than 60 peptides as medications, while more than 500 therapeutic peptides are undergoing preclinical testing [10,11]. Several examples of short, potent peptides that have been used successfully in clinical practice include 9-aa (amino acid) gonadotropin receptor agonists (lupron/leuprorelin and zoladex/goserelin for breast and prostate cancer), 2-aa proteasome inhibitors (velcade/bortezomib for multiple myeloma), and 10-aa immunomodulators (copaxone/glatiramer acetate for multiple sclerosis) [11]. Current strategies for combating β-lactamase resistance focus on the use of smaller molecular inhibitors or more recent β-lactam antibiotics. Regarding the use of peptides for combating β-lactamase resistance, the antimicrobial peptide thanatin was shown to be effective against both extended spectrum β-lactamases (ESBLs) and New Delhi metallo β-lactamases (NDMs) producing bacteria. Similarly, the peptides nisin, plectasin, and dBLIP-1 and -2 have been explored as β-lactamase inhibitors or β-lactam adjuvants [12,13,14,15,16]. The current study aimed to evaluate the antimicrobial efficacy of ten novel pentapeptides obtained after a systematic screening of a library of peptides through in silico approaches and cellular and in vitro assays. The goal of this study was to identify novel peptide inhibitors that might act as potentiators of β-lactam activity in existing drug-resistant strains. Traditional (ESBLs), such as TEM-1 and SHV-14, are prevalent all over the world and play a significant role in governing drug resistance [4]. Due to their abundance in nature, we chose to use TEM-1 and SHV-14 to gain preliminary insight into the action of the peptides.

## 2. Materials and Methods

### 2.1. Bacterial Strains and Plasmids for Cellular and Molecular Studies

The bacterial strains and plasmids used for this study are shown in Table 1. The β-lactamase genes (*bla*_SHV-14_ and *bla*_TEM-1_) were amplified through PCR from the chromosomal DNA of *Klebsiella pneumoniae* type strain MTCC 3384 and a clinical strain of *Escherichia coli*, respectively. The full-length constructs (*bla*_SHV-14_ and *bla*_TEM-1_) were cloned and expressed in pBAD18-Cam (*Cam^R^*) vector under an arabinose-inducible promoter and in the pET28a(+) (*Kan^R^*) vector under a *lac*-inducible promoter to conduct the cellular and in vitro studies, respectively [4]. *E. coli* BL21 (DE3) pLysS strain was used for the expression and purification of proteins, while the CS109 (lacking O-antigens) strain was used to assess cellular activity [17] (a generous gift from Professor Waldemar Vollmer, Newcastle University, UK).

### 2.2. Media, Antibiotics, Peptides, and Culture Conditions for Biological Assays

All of the reagents, including antibiotics and molecular biology reagents utilised in this investigation, were purchased from Sigma-Aldrich (St. Louis, MO, USA) and New England Biolabs (Ipswich, MA, USA), respectively. Antibiotic stock solutions were made by dissolving the antibiotic powder in sterile water, DMSO, or ethanol by either volume/volume (*v*/*v*) or weight/volume (*w*/*v*) according to the manufacturers’ instructions. For peptide stock solutions, 0.01 g of each peptide was dissolved in 1 mL of nuclease-free water to make stocks of 10 mg/mL, and the working concentration was 100 µg/mL, which was used for the biological analyses.

### 2.3. In Silico Modelling and Peptide Library Construction

Primarily, a group of peptide sequences (n = 102) was designed after a detailed analysis of the Ω-loop sequences of various class A β-lactamases (TEM, SHV, CTX-M). Considering the Ω-loop sequence as a template, a group of linear pentapeptides was created. The pentapeptides were principally constructed on the basis of charge complementarity, allowing for strong electrostatic interactions with the amino acids of the Ω-loop. Additionally, factors including peptide length, solubility (the influence of hydrophobic and hydrophilic amino acids on solubility), hydropathy (charged/uncharged residues), and intermolecular H-bond forming capacity were analysed in detail before considering individual amino acids for the peptide sequences (Appendix A). The peptide sequences were taken for 3D model generation (Figure 1) via the peptide structure prediction server module (Biopredicta) of V-life MDS [20]. PDB formats of the peptides were considered for further docking investigations.

### 2.4. Molecular Docking of Peptides into the Active Site Pocket of Class A β-Lactamases

The generated peptide models were subjected to in silico docking via the AutoDock4.2 tool with a class A β-lactamase (3D-dimensional structure of SHV-14) to analyse potential binding interactions [4,21]. The grid box was generated in a centre grid space of x = 0.191 Å, y = 29.778 Å, z = 11.574 Å with offset values of x = −6.111 Å, y = −29.806 Å and z = −2.722 Å. The efficacy value, free energy, and RMSD between conformations were used to evaluate the molecular docking. The docking results were ranked by lower binding energy (ΔG_binding_).

### 2.5. Chemical Synthesis of Peptides by Solid-Phase Peptide Synthesis (SPPS)

Peptides were synthesised simultaneously using automated Fmoc SPPS on a Biotage Syro I automated parallel peptide synthesiser [22] using 100 mg of either H-His(Trt)-2-ClTrt resin, loading 0.71 mmol g^−1^, or Fmoc-Arg(Pbf)-Wang resin, loading 0.59 mmol g^−1^. The Fmoc amino acids used were: Fmoc-Glu (O*t*Bu)-OH, Fmoc-His(Trt)-OH, Fmoc-Ile-OH, Fmoc-Leu-OH, Fmoc-Arg(Pbf)-OH, Fmoc-Ser(*t*Bu)-OH, Fmoc-Thr(*t*Bu)-OH, and Fmoc-Tyr(*t*Bu)-OH. Peptides were dissolved in the required amount of HCl and lyophilized to produce the final product (method details and characterisation data provided as Appendix A).

### 2.6. High-Performance Liquid Chromatography (HPLC) and High-Resolution Mass Spectroscopy (HRMS)

Peptides were purified by preparative reverse phase HPLC on an Agilent Infinity 1260 Prep system with an Infinity II fraction collector, using an Agilent ZORBAX 300SB-C18 7 µm 21.2 × 250 mm column and detection at 214 nm. A linear solvent gradient of 25 to 40% MeCN (0.1% TFA) in H_2_O (0.1% TFA) over 7.5 min was used at a flow rate of 20 mL min^−1^. Accurate mass spectra were recorded on an Agilent LC system connected to an Agilent 6510 Q-TOF mass spectrometer.

### 2.7. Protein Purification

The N-terminal 6X histidine-tagged proteins (SHV-14 and TEM-1) were expressed from the respective plasmid constructs already available in the laboratory and were purified by Ni-NTA affinity chromatography using an AKTA prime plus (GE Healthcare, Piscataway, NJ, USA) as described previously [4]. Nitrocefin hydrolysis was used to validate the β-lactamase activity of both proteins.

### 2.8. Antibiotic Susceptibility Testing

#### 2.8.1. Whole-Cell Phenotypic Evaluation of Synthesized Peptides by HT-SPOTi

HT-SPOTi (high-throughput spot culture growth inhibition) assay was used to evaluate the whole cell phenotypic effect of peptides against a selection of Gram-negative and acid-fast bacterial strains in the presence of β-lactam antibiotics as previously described [23]. The distribution of the molten agar media into the 96 wells was achieved using the Multidrop Combi microplate dispenser (Thermo-Fisher Scientific). Plates were left for 5 min for the agar to set, and a diluted inoculum of bacteria was applied (2 µL of ∼10^3^ CFU) into each well by either the use of a multi-channel pipettor or by using the Multidrop Combi [24]. DMSO was taken as no antibiotic control and kanamycin as non-β-lactam drug control. Unless otherwise specified, all the experiments were performed in triplicate.

#### 2.8.2. Determination of MIC Using Broth Microdilution Method

Antibiotic susceptibility was assessed by determining minimum inhibitory concentration (MIC) using the broth microdilution method as per CLSI guidelines and described previously [25,26]. The antibiotics were serially diluted in a 96-well microtiter plate, and the volume was adjusted to 300 μL per well with MHB. A total of 2 μL (100 µg/mL) of each peptide was added to every well, and finally, the bacterial cultures were added in such a way that each inoculum contained ∼10^5^ cells/well. The MICs were determined by observing the OD at 600 nm using a Multiskan Spectrum spectrophotometer (model 1500; Thermo Scientific, Nyon, Switzerland). Unless otherwise specified, all the experiments were performed in triplicate.

### 2.9. Kinetic Behaviour of β-Lactamases in the Presence of the Peptides

Assessment of enzyme activities was carried out by calculating the kinetic parameters [Michaelis–Menten constants (*K_m_* and *V_max_*), turn-over number (*k_cat_*), and catalytic efficiency (*k_cat_*/*K_m_*)] with various β-lactam substrates in the presence of the peptides as described previously [4,27]. The hydrolysis reaction was carried out at 25 °C in 10 mM Tris-HCl, 300 mM NaCl (pH 7.5) buffer supplemented with 1 μg mL^−1^ bovine serum albumin for protein stability. The enzyme concentrations (0.46 μM: SHV-14 and 0.42 μM: TEM-1) were taken to determine the initial rate constants at varying substrate concentrations (20–500 μM). The values were calculated using the online curve-fitting tool (https://www.mycurvefit.com, MyAssays Ltd., Brighton, UK, accessed on 25 May 2022).

### 2.10. Molecular Dynamics Simulation Study

MD simulation was performed using the AMBER99SB-ILDN force field in the GROMACS 5.0.4 package [28,29] for a 100 ns time scale for peptides (P5, P6, P7, P8, P9, and P10). TIP3P force field was used to build the water model (dodecahedron box with 10 Å of edge distance) [30,31,32,33] (refer to Appendix A for details). The dynamics stability parameters (fluctuations and conformational changes) of the peptides were investigated through the root mean square deviation (RMSD), radius of gyration (Rg), and root mean square fluctuation (Cα-RMSF) analysis. 

## 3. Results and Discussion

### 3.1. Docking Analysis Supported the Hypothesis That Peptides Could Inhibit β-Lactamase

The mode of inhibition of pentapeptides on β-lactamases was evaluated by molecular docking analysis. The distance calculations indicated that specific amino acids of the ῼ-loop (Asp-159, Arg-160, Trp-161, Glu-162, Thr-163, and Glu-164) played a crucial role in tightly anchoring the peptides within the binding pocket via strong hydrogen bonds (Figure 2). The presence of non-polar, hydrophobic residues such as Ile, Leu, and Tyr in the peptides enhanced the hydrophobic interactions with the loop, thus strengthening the binding. Therefore, a dense network of intermolecular contacts (hydrogen bonds and hydrophobic and electrostatic interactions) clearly demonstrated strong binding of the peptides to the ῼ-loop. Ten peptides carrying five amino acids each [EYRIR (P1), TYRLR(P2), TSHLR (P3), TTHIR (P4), ETHIH (P5), TSHLH (P6), ESRLH (P7), ESHIH (P8), ESRIH (P9), TYHLH (P10)] showing the highest affinity and most energetically favoured structures (Table 2), were chosen for SPPS. The purity and identity of the peptides were validated using reverse-phase HPLC and high-resolution MS. However, we only managed to achieve less than 95% purity for peptides 1–4.

### 3.2. The Antimicrobial Efficacy of the β-Lactams Was Enhanced in the Presence of the Peptides in Gram-Negative Organisms

To check the inhibitory activity of peptides and comprehend particular structure–activity relationships, HT-SPOTi assay was performed. The whole-cell phenotypic effect of the 10 pentapeptides was ascertained in combination with β-lactam antibiotics in *E. coli* CS109 and *K. pneumoniae*.

*E. coli* CS109 strains generally displayed lower MICs for all the penicillin group of antibiotics tested in the presence of peptides than that of the control (P5: ~2–8 fold, P6: ~2–4 fold, P7: ~2–4 fold, P8: ~2–4 fold, P9: ~2–4 fold, P10: ~2–4 fold) (Table 3a). Furthermore, several peptides exhibited significant antibacterial activity against bacteria in combination with β-lactam drugs in *K. pneumoniae*. The sensitivity of cells to the β-lactam drugs was increased significantly with the peptides, *viz*., P5: ~2–4 fold, P6: ~2–8 fold, P7: ~2–4 fold, P8: ~2–4 fold, P9: ~2–8 fold, P10: ~2–8 fold (Table 3b). However, the levels of resistance to the cephalosporin compounds tested (cefadroxil, cefaclor, cefoperazone, ceftazidime, and cefepime) remained unaltered with peptides. The result led us to believe that in the case of Gram-negative strains, the co-treatment with the synthesized peptides and β-lactams demonstrated a considerable shift in the resistance profile.

The β-lactamases are secretory in nature and normally found in the periplasmic spaces of Gram-negative bacteria. It has been previously reported that cationic pentapeptides can penetrate cells by crossing the plasma membrane [2,34]. Antimicrobial peptides with positive charges are attracted to polyanionic outer surfaces (due to cell-wall-associated teichoic and lipoteichoic acids) in Gram-positive bacteria and lipopolysaccharides (LPS) in Gram-negative bacteria [34]. The presence of positive charges then leads to the electrostatic interaction of peptides with the phospholipid bilayer, which improves the entry of peptides into the cell by disrupting the membrane [35]. As all the peptides in our study are cationic pentapeptides, membrane disruption could be a contributing factor to their antimicrobial activity.

### 3.3. Enhanced Antimicrobial Action of Peptides in Mycobacteria

It is apparent that the glutamic acid in the ῼ-like loop of class A β-lactamases (MSMEG_2433 and MSMEG_4455) considerably impacts β-lactamase activity in *Mycobacterium smegmatis* [5,36]. To check whether the pentapeptides designed for this study have any influence on mycobacterial β-lactam resistance, we tested for bacterial susceptibility towards β-lactam drugs in the presence and absence of the pentapeptides in *M. smegmatis* and *M. tuberculosis* (H37Rv strain and MDR-clinical isolates). It was observed that the combination of peptides and β-lactams sensitised the *M. smegmatis* strain by making them more vulnerable to β-lactam drugs (Table 3c). Furthermore, substantial differences in antibiotic susceptibility were observed for penicillin groups with peptides than that of the control without peptides (P5: ~2–4-fold, P6: ~2–8 fold, P7: ~2–4 fold, and P8: ~2–4 fold), except for piperacillin and cephalosporins. Tests employing peptides in combination with ampicillin on *M. tuberculosis* H37Rv laboratory strain and MDR-TB clinical isolates revealed that the combination was more effective in the case of *M. tuberculosis* H37Rv. To our surprise, the MICs in H37Rv revealed a drastic enhancement of susceptibility towards ampicillin upon the introduction of peptides (P5: ~8-fold, P6: ~32-fold, P7: ~16-fold, and P8: ~16-fold) (Appendix A). Similarly, our peptides were tested in combination with ampicillin against *M. tuberculosis* clinical isolate (MDR-Peru isolate). These results indicated that certain peptides appear to have a profound effect on antibiotic resistance, increasing ampicillin susceptibility by 2–32-fold; in other words, the loss of sensitivity towards these drugs was recovered in the presence of some peptides. In contrast, the MIC values of the cells treated with only peptides varied from 64–256 µg/mL for E. coli, >500 µg/mL for M. smegmatis, and >256 µg/mL for K. pneumoniae (Appendix A), which showed the sensitization of the respective bacterial cells upon treatment with the respective peptides in combination with β-lactams.

Earlier research revealed that broad-spectrum class A β-lactamases (Amber classification) are responsible for more than 80% of the overall β-lactamase activity in *M. tuberculosis* [37,38]. According to Voladri et al., the β-lactamases in *M. tuberculosis* are predominantly penicillinases and have a conserved sequence comparable to the ῼ-loop of TEM-1 and SHV-14 [39,40]. Similarly, BlaA and BlaC share homology with the major β-lactamases from *M. smegmatis* mc^2^155, which has some similarities with BlaF, the well-studied class A β-lactamase [41]. Regardless of whether they are encoded by chromosomes or plasmids, class A β-lactamases share identical domains or folds. Nonetheless, all the aforementioned evidence indicates that the peptides are sensitizing the mycobacterial cells to β-lactam agents, possibly by inhibiting the class A β-lactamases.

Therefore, our research led us to the possibility that the six histidine-based pentapeptides (TSHLH, ETHIH, ESRLH, ESHIH, ESRIH, and TYHLH) might act as β-lactamase inhibitors, reducing the MICs and significantly affecting the resistance profile of β-lactams in various bacteria. However, the change in MIC value due to the arginine-based peptides was modest, as demonstrated by the minimal decrease in MIC value. Therefore, to further validate our hypothesis, we prioritised the six histidine-based peptides and ruled out the arginine-based peptides.

### 3.4. Cells Expressing the β-Lactamases Confirmed Enhanced Sensitivity with Peptides

Many antimicrobial peptides kill bacteria by physically interacting with cell membranes [42]. However, Hale and Hanlock revealed that some cationic peptides which lack a structured conformation are membrane-non-disruptive and potentially have many non-membrane targets owing to their amphiphilic nature, and most bacteria are potentially susceptible to cationic antimicrobial peptides [43,44]. It was therefore crucial for us to ascertain whether the peptides could traverse the bacterial membranes and bind to the β-lactamases. Two of the most common β-lactamase enzymes that Gram-negative bacteria use to overcome β-lactam antibiotics are class A beta-lactamases, namely TEM-1 and SHV-14. The Ω-loop residues of TEM-1 and SHV-14 β-lactamases were also used for the design of the peptides in this study. Hence, we evaluated the antimicrobial activity of the six histidine-based peptides in the *E. coli* CS109 strain upon individually overexpressing two β-lactamase genes (*bla*_TEM-1_ and *bla*_SHV-14_) through HT-SPOTi assay. The cloned *bla*_TEM-1_ and *bla*_SHV-14_ in pBAD18cm were ectopically expressed in *E. coli* CS109 cells using arabinose as the inducer. The cells expressing *bla*_TEM-1_ became sensitive towards all the β-lactams tested when treated with peptides P5, P6, P7, P8, P9, and P10 (penicillin: ~2–4 fold, ampicillin: ~2–8 fold, amoxicillin: ~2–8 fold, piperacillin: ~8 fold) as compared to the controls without peptides (Table 4). Similarly, the presence of the peptides significantly decreased the MICs for the cells expressing *bla*_SHV-14_ (penicillin: ~2–4 fold, ampicillin: ~2–8 fold, amoxicillin: ~4–8 fold, piperacillin: ~2–8 fold) (Table 4). 

Peptides P5–P10 exhibited a similar impact on the MIC of β-lactams tested when the experiments were conducted using the broth microdilution method. The MICs of all the tested antibiotics decreased dramatically in the presence of the peptides upon in trans expression of β-lactamases. The ectopic expression of *bla*_TEM-1_, when combined with the peptides, displayed a drastic reduction in β-lactam resistance (penicillin: ~2–4 fold, ampicillin: ~2–8 fold, amoxicillin: ~2–8 fold, piperacillin: ~8 fold) as compared to the control cells (expressing *bla*_TEM-1_) treated only with the β-lactams. Similarly, a considerable increase in susceptibility of bacteria expressing *bla*_SHV-14_ was observed towards β-lactams upon treatment with the peptides, viz., penicillin: ~2–4 fold, ampicillin: ~4–8 fold, amoxicillin: ~2–4 fold, and piperacillin: ~2–8 fold (Table 5).

These findings indicated that use of β-lactam antibiotics in combination with peptides P5-P10 is effective. The peptides suppress the in vivo function of β-lactamases, which in turn reduces the degradation of β-lactams, resulting in restored sensitivity towards these drugs in strains expressing class A β-lactamases. Overall, we can infer that the peptides are inhibiting class A β-lactamases and therefore restoring the activity of β-lactams against strains expressing these enzymes.

### 3.5. Peptides Have a Significant Inhibitory Impact on the Kinetic Behaviour of β-Lactamases

To understand whether the kinetic behaviour of the enzymes is in synchrony with the cellular susceptibility data (MIC), kinetic parameters for β-lactam substrates were determined for TEM-1 and SHV-14. Due to their broad-spectrum catalytic profiles, both enzymes showed good catalytic efficiency (*k_cat_*/*K_m_*), but the hydrolysis was minimal with peptides. The enzyme affinity (*K_m_*) value was increased by more than 10 times with peptides for both enzymes, which is an indication of reduction in the binding affinity. Similarly, a sharp decline in turnover number in the presence of peptides was observed as compared to the proteins alone (TEM-1 *k_cat_* > 290 s^−1^, SHV-14 *k_cat_* > 400 s^−1^) (Table 6a,b). All of the aforementioned data indicated that peptides hinder the effectiveness of the β-lactamases by modifying the overall enzymatic efficiency. According to Rudgers, SHV-1 and TEM-1 β-lactamases share 68% structural identity with each other [34]. Multiple sequence alignment showed that SHV-14 and TEM-1 also have 80% sequence and structural similarity. Due to the close resemblance of the SHV-14 and TEM-1 active-site pockets, it was anticipated that peptides would have similar affinities for both enzymes. Additionally, it is reported that the substrate profile of TEM-1 and SHV-1 are similar [34]. The peptides used here possibly target the expressed β-lactamases to nullify their activity irrespective of the bacterial strains that are harbouring the enzymes. We can assume that the similarity between SHV-14 and TEM-1 explains their similar hydrolytic capacity in vivo and in vitro, and their similar levels of suppression by the peptides. According to the literature, although the proteins readily hydrolyse the penicillin group of antibiotics, neither SHV-14 nor TEM-1 are able to hydrolyse cephalosporins [34]. Accordingly, none of the synthesised peptides affected the MICs of any of the cephalosporins tested.

### 3.6. MD Simulation Supports the Stability of Peptides

The six pentapeptides were identified as bearing secondary structures matching with the Hale and Hanlock classification of loop-type secondary structures [43,44]. The backbone RMSDs were determined for each to observe their folding pattern (Figure 3), which reflected their stability after ~30 ns (P7A), ~60 ns (P10A), ~80 ns (P6A), and between ~90–100 ns (P5A, P8A, P9A). The RMSFs of Cα atoms were calculated with minor residue-wise fluctuation within the range of ~0.1–0.2 nm (P6B, P8B, P10B), ~0.12–0.2 nm (P5B), ~0.06–0.12 nm (P7B), and ~0.1–0.25 nm (P9B). Structure-wise stability was studied from the Rg plots, which indicated that all six peptide structures (P5C-P10C) achieved good compactness within 100 ns of MD simulation. Interestingly, P6 (P6C) and P7 (P7C) peptides achieved structural stability more rapidly than the other peptides, i.e., after ~30 ns and ~15 ns, respectively. Further, better conservation of hydrogen bonds was observed in the case of P6, P7, P8, and P9 compared to P5 and P10 throughout the simulation period of 100 ns (P5D-P10D). Our RMSF analyses revealed that the P5 (P5B) and P7 (P7B) peptides appeared to have much lower residue-wise fluctuation levels than the other peptides (P6, P8, P9, and P10). The folding patterns of P7 and P10 also reached stability more quickly than those of the other four peptides. Based on the intrinsic stability and stable evolution of secondary structure elements during the MD simulation, all six pentapeptides achieved structural stability within a 100 ns time frame.

## 4. Conclusions

Overall, the present study investigated the inhibitory activity of ten synthetic pentapeptides in several bacteria, including cells harbouring the ectopically expressed β-lactamase. Among the ten pentapeptides, histidine-based peptides (TSHLH, ETHIH, ESRLH, ESHIH, ESRIH, and TYHLH) were found to display bactericidal activities against a panel of Gram-negative and acid-fast bacteria in combination with all the antibiotics tested. Based on the obtained results, the six histidine-based pentapeptides exhibited inhibitory activity over the ectopically expressed TEM-1 and SHV-14 β-lactamases. Our computational methods supported the experimental results and revealed that bacterial inhibitory peptides might be used as candidates for future antimicrobial chemotherapy. However, the unique ability of peptides to inhibit class A β-lactamases opens up a promising new direction for the development of novel inhibitors to treat β-lactamase drug resistance. Nevertheless, further studies are needed to confirm the exact molecular mechanism of the peptides with the Ω-loop of the class A β-lactamases, including the investigation of peptide cyclization, isothermal titration calorimetry to confirm β-lactamase binding, and cryo-electron microscopy to confirm peptide localisation. The results of such experiments are expected to enable identification of the most promising lead from among the six shortlisted peptides.

## Figures and Tables

**Figure 1 antibiotics-12-00553-f001:**
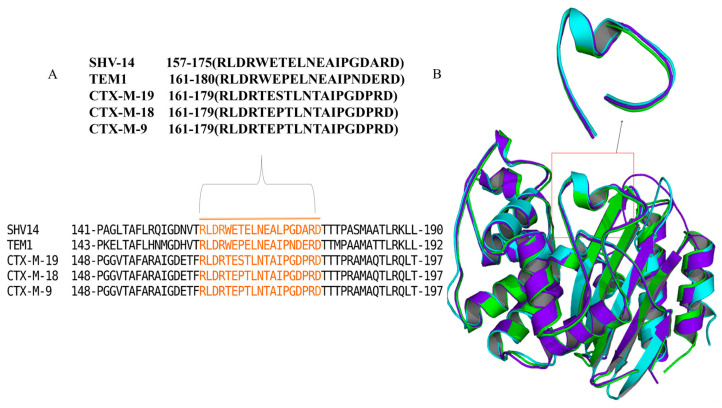
(**A**) Multiple sequence alignment between the class A β-lactamases, focusing on the conserved sequence of the Ω-loop. Residues highlighted in orange indicate sequences of the Ω-loop. (**B**) Schematic diagram of backbone superimposition between the enzymes; TEM-1 (blue), SHV-14 (cyan), and CTX-M (green). The focused section shows a close-up view of the Ω-loop.

**Figure 2 antibiotics-12-00553-f002:**
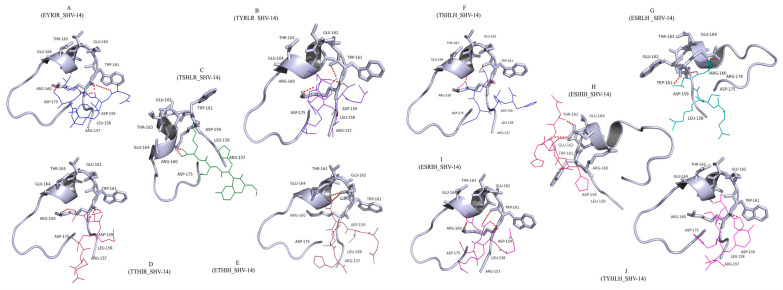
Schematic representations of docking analysis and interatomic interaction study of the protein SHV-14 with peptides. The orientations of all the peptides in the active site after docking: (**A**) P1_SHV-14 docking complex, (**B**) P2_SHV-14 docking complex, (**C**) P3_SHV-14 docking complex, (**D**) P4_SHV-14 docking complex, (**E**) P5_SHV-14 docking complex, (**F**) P6_SHV-14 docking complex, (**G**) P7_SHV-14 docking complex, (**H**) P8_SHV-14 docking complex, (**I**) P9_SHV-14 docking complex, and (**J**) P10_SHV-14 docking complex. The protein is represented in cartoon mode, the key active site residues are depicted in line representation, and all the peptides are indicated in stick representation.

**Figure 3 antibiotics-12-00553-f003:**
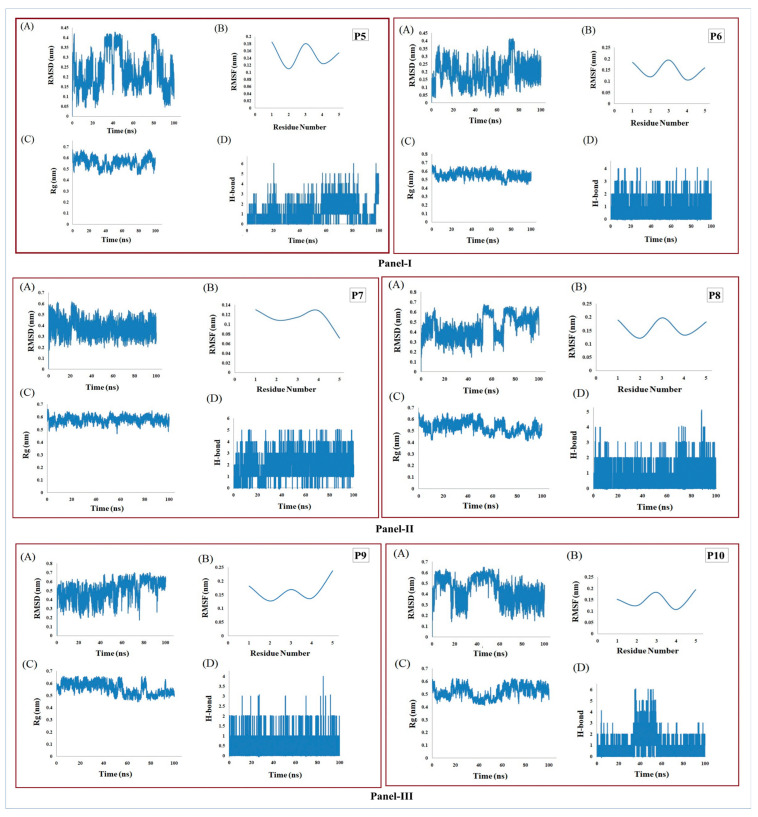
Study of conformational stability of pentapeptides via MD simulations and their trajectories represented as Panel I: P5 and P6, Panel II: P7 and P8, Panel III: P9 and P10. Each panel contains (**A**) RMSD analysis of peptides, (**B**) comparative RMSF of Cα atoms, (**C**) radius of gyration plot, (**D**) variation of hydrogen bonds participating in the intermolecular interactions.

**Table 1 antibiotics-12-00553-t001:** Bacterial strains and plasmids.

Strains/Plasmids	Source or Reference	
*E. coli* CS109	C. Schnaitman; Ghosh and Young, 2005 [18]	
*S. aureus*	ATCC25923	
*Klebsiella pneumoniae*	NCTC Number: 12463	
*Klebsiella pneumoniae*	MTCC 3384 (Kumar et al., 2018) [4]	
*M. tuberculosis* MDR strain	Peru isolate (Grandjean et al., 2008) [19]	
*M. tuberculosis* H37Rv	ATCC25923	
*M. smegmatis* mc^2^155	ATCC700084	
pBAD18Cam	Expression vector with arabinose-inducible promoter	Stratagene
pET28a(+)	*E. coli* expression vector generating His6 fusion proteins for overexpression	Stratagene
BL21 (DE3) pLysS	F–, ompT, *hsd*SB (rB –, mB –), *dcm*, *gal*, λ(DE3), pLysS (Cmr)	Promega

**Table 2 antibiotics-12-00553-t002:** Docking results of peptides along with their binding energy (ΔG_binding_).

Peptide Sequence	Binding Energy kcal/mol	Hydrogen Bonds	Interacting Amino Acids
EYRIR (P1)	−3.28	3	Arg160, Trp161 (2)
TYRLR(P2)	−6.56	3	Arg160 (2), Trp161
TSHLR (P3)	−5.88	3	Arg160, Trp161
TTHIR (P4)	−4.56	3	Leu158, Asp159, Asp175
ETHIH (P5)	−3.42	4	Arg160 (3), Trp161
TSHLH (P6)	−4.68	5	Arg160 (4), Trp161,
ESRLH (P7)	−5.84	5	Arg160 (2),Trp161, Glu164
ESHIH (P8)	−4.25	4	Arg160 (3), Trp161
ESRIH (P9)	−3.68	4	Arg160 (3), Trp161
TYHLH (P10)	−6.78	3	Trp161, Glu162, Thr163

**Table 3 antibiotics-12-00553-t003:** MIC values of β-lactam drugs in presence of peptides through HT-SPOTi. (**a**) *E. coli* strain CS109. (**b**) K. pneumoniae strain. (**c**) M. smegmatis MC2 155.

(**a**)				
**Strain**	**MIC Value Antibiotics (mg/L)**			
	**Pen**	**Amp**	**Amx**	**Pip**
*E. coli* CS109	**32**	**4**	**8**	**16**
*E. coli* CS109 + P1	16	4	4	16
*E. coli* CS109 + P2	16	2	8	4
*E. coli* CS109 + P3	32	8	8	8
*E. coli* CS109 + P4	16	4	4	16
*E. coli* CS109 + P5	32	2	2	2
*E. coli* CS109 + P6	16	1	2	8
*E. coli* CS109 + P7	16	4	2	4
*E. coli* CS109 + P8	32	4	4	4
*E. coli* CS109 + P9	16	2	8	4
*E. coli* CS109 + P10	16	2	8	4
DMSO	>500	>500	>500	>500
Kanamycin	64	64	64	>500
(**b**)				
**Strain**	**MIC Value Antibiotics (mg/L)**			
	**Pen**	**Amp**	**Amx**	**Pip**
*K. pneumoniae*	**32**	**32**	**64**	**64**
*K. pneumoniae +* P5	8	8	16	32
*K. pneumoniae +* P6	8	4	32	16
*K. pneumoniae +* P7	16	8	16	16
*K. pneumoniae +* P8	8	8	16	16
*K. pneumoniae +* P9	8	4	32	16
*K. pneumoniae +* P10	8	4	32	32
DMSO	>500	>500	>500	>500
Kanamycin	64	64	64	64
(**c**)				
**Strain**	**MIC Value Antibiotics (mg/L)**			
	**Pen**	**Amp**	**Amx**	**Pip**
*M. smegmatis*	**125**	**4**	**2**	>250
*M. smegmatis +* P1	125	4	2	>250
*M. smegmatis +* P2	64	4	2	>250
*M. smegmatis +* P3	125	8	8	>250
*M. smegmatis +* P4	64	8	4	>250
*M. smegmatis +* P5	32	1	1	>250
*M. smegmatis +* P6	32	0.5	0.5	>250
*M. smegmatis +* P7	64	1	1	>250
*M. smegmatis +* P8	32	4	1	>250
*M. smegmatis +* P9	32	2	2	>250
*M. smegmatis +* P10	64	2	2	>250
DMSO	>500	>500	>500	>250
Isoniazid	64	64	64	>250

Footnotes: Pen: Penicillin, Amp: Ampicillin, Amx: Amoxicillin, Pip: Piperacillin.

**Table 4 antibiotics-12-00553-t004:** MICs of β-lactam drugs against *Escherichia coli* CS109 harbouring TEM-1 and SHV-14 determined through HT-SPOTi in presence of peptides.

Strain	MIC Value of Antibiotics (mg/L)			
	Pen	Amp	Amx	Pip
*E. coil* CS109	32	4	8	16
*E. coli* **CS109 + pBAD-TEM1**	**64**	**16**	**32**	**64**
*E. coil* CS109 + pBAD-TEM1 + P5	16	4	8	8
*E. coil* CS109 + pBAD-TEM1 + P6	16	2	16	8
*E. coil* CS109 + pBAD-TEM1 + P7	16	4	4	8
*E. coil* CS109 + pBAD-TEM1 + P8	32	4	16	8
*E. coil* CS109 + pBAD-TEM1 + P9	16	8	8	8
*E. coil* CS109 + pBAD-TEM1 + P10	16	8	8	8
*E. coli* **CS109 + pBAD-SHV-14**	**250**	**32**	**125**	**64**
*E. coli* CS109 + pBAD-SHV-14 + P5	64	32	16	16
*E. coli* CS109 + pBAD-SHV-14 + P6	125	4	16	8
*E. coli* CS109 + pBAD-SHV-14 + P7	64	4	32	16
*E. coli* CS109 + pBAD-SHV-14 + P8	32	4	32	32
*E. coli* CS109 + pBAD-SHV-14 + P9	125	8	32	32
*E. coli* CS109 + pBAD-SHV-14 + P10	32	4	32	32
DMSO	>500	>500	>500	>500
Kanamycin	64	64	64	64

Footnotes: Pen: Penicillin, Amp: Ampicillin, Amx: Amoxicillin, Pip: Piperacillin.

**Table 5 antibiotics-12-00553-t005:** MICs of β-lactam drugs against *Escherichia coli* CS109 harbouring TEM-1 and SHV-14 determined through broth microdilution method in presence of peptides.

Strain	MIC Value of Antibiotics (mg/L)			
	Pen	Amp	Amx	Pip
*E. coil* CS109	64	8	16	16
*E. coli* **CS109 + pBAD-TEM1**	**125**	**32**	**64**	**125**
*E. coil* CS109 + pBAD-TEM1 + P5	32	8	16	16
*E. coil* CS109 + pBAD-TEM1 + P6	32	4	32	16
*E. coil* CS109 + pBAD-TEM1 + P7	32	8	8	16
*E. coil* CS109 + pBAD-TEM1 + P8	64	8	16	16
*E. coil* CS109 + pBAD-TEM1 + P9	32	16	16	16
*E. coil* CS109 + pBAD-TEM1 + P10	32	16	16	16
*E. coli* **CS109 + pBAD-SHV-14**	**500**	**64**	**125**	**125**
*E. coli* CS109 + pBAD-SHV-14 + P5	125	64	32	32
*E. coli* CS109 + pBAD-SHV-14 + P6	250	8	32	16
*E. coli* CS109 + pBAD-SHV-14 + P7	125	8	64	32
*E. coli* CS109 + pBAD-SHV-14 + P8	250	8	64	64
*E. coli* CS109 + pBAD-SHV-14 + P9	250	16	64	64
*E. coli* CS109 + pBAD-SHV-14 + P10	250	8	64	64

Footnotes: Pen: Penicillin, Amp: Ampicillin, Amx: Amoxicillin, Pip: Piperacillin.

**Table 6 antibiotics-12-00553-t006:** Kinetic parameters of (**a**) TEM-1 and (**b**) SHV-14 with β-lactam antibiotics and peptides.

(**a**)			
**TEM-1**	***K*_m_ (μM)**	***k_cat_* (s^−1^)**	***k_cat_*/*K*_m_ (s^−1^μM^−1^)**
Pen	67.20 ± 4.43	386.95 ± 15.97	5.4
Pen + P5	504.34 ± 72.56	14.89 ± 0.74	0.027
Pen + P6	331.78 ± 29.30	12.28 ± 1.87	0.036
Pen + P7	324.04 ± 20.20	11.42 ± 1.52	0.033
Pen + P8	290.48 ± 20.81	31.1 ± 1.46	0.106
Pen + P9	281.01 ± 47.47	19.03 ± 2.5	0.67
Pen + P10	240.38 ± 50.54	30.05 ± 0.5	0.125
**TEM-1**	***K*_m_ (μM)**	***k_cat_* (s^−1^)**	***k_cat_*/*K*_m_ (μM)**
Amp	66.04 ± 12.16	290 ± 85.55	4.3
Amp + P5	442.54 ± 37.75	23.43 ± 1.09	0.052
Amp + P6	358.03± 47.30	17.33 ± 0.03	0.047
Amp + P7	602.09 ± 26.36	5.99 ± 0.05	0.009
Amp + P8	328.99 ± 67.04	14.56 ± 2.03	0.042
Amp + P9	289.40 ± 31.05	24.12 ± 0.87	0.081
Amp + P10	179.67 ± 30.07	51.08 ± 1.44	0.28
**TEM-1**	***K*_m_ (μM)**	***k_cat_* (s^−1^)**	***k_cat_*/*K*_m_ (μM)**
Amx	105.09 ± 12.16	427 ± 20.99	4
Amx + P5	410.62 ± 102.46	31.91 ± 6.27	0.07
Amx + P6	262.35 ± 93.87	20 ± 2.02	0.70
Amx + P7	668.78 ± 59.91	14.29 ± 1.46	0.02
Amx + P8	452.78 ± 45.01	23.01 ± 0.87	0.05
Amx + P9	342.45 ± 23.11	2.1 ± 0.09	0.005
Amx + P10	487.32 ± 14.21	14.34 ± 3.02	0.02
**TEM-1**	***K*_m_ (μM)**	***k_cat_* (s^−1^)**	***k_cat_* /*K*_m_ (μM)**
Pip	136.32 ± 12.16	331.26 ± 12.22	2.4
Pip + P5	787.11 ± 122.49	14.25 ± 1.09	0.01
Pip + P6	521.08 ± 34.79	23.9 ± 1.41	0.044
Pip + P7	387.78 ± 129.8	32.01± 6.43	0.023
Pip + P8	737.46 ± 67.98	12.12± 4.11	0.016
Pip + P9	814.67 ± 33.19	27.32 ± 5.56	0.033
Pip + P10	768.09 ± 78.66	19.09 ± 3.03	0.024
(**b**)			
**SHV-14**	***K*_m_ (μM)**	***k_cat_* (s^−1^)**	***k_cat_*/*K*_m_ (s^−1^μM^−1^)**
Pen	144.20 ± 4.43	586.95 ± 15.97	4.07
Pen + P5	343.98 ± 19.94	42.32 ± 3.01	0.12
Pen + P6	453.05 ± 8.97	18.14 ± 2.32	0.04
Pen + P7	564.01 ± 6.89	15.2 ± 1.21	0.02
Pen + P8	612.34 ± 11.32	34.13 ± 3.12	0.05
Pen + P9	432.28 ± 32.9	38.81 ± 0.1	0.08
Pen + P10	461.32 ± 22.01	11.24 ± 2.1	0.02
**SHV-14**	***K*_m_ (μM)**	***k_cat_* (s^−1^)**	***k_cat_* /*K*_m_ (μM)**
Amp	186.74 ± 12.16	644.95 ± 10.97	3.4
Amp + P5	603.42 ± 7.02	14.11 ± 1.03	0.02
Amp + P6	560.99 ± 43.12	19.5 ± 2.05	0.03
Amp + P7	667.51 ± 16.43	12.09 ± 1.01	0.01
Amp + P8	340.21 ± 6.86	54.7 ± 0.09	0.1
Amp + P9	365.34 ± 35.1	34.43 ± 0.98	0.07
Amp + P10	276.56 ± 34.09	34.03 ± 4.33	0.01
**SHV-14**	***K*_m_ (μM)**	***k_cat_* (s^−1^)**	***k_cat_*/*K*_m_ (μM)**
Amx	111.74 ± 12.16	416.95 ± 5.02	3.7
Amx + P5	626.02 ± 24.01	23.02 ± 0.04	0.03
Amx + P6	564.55 ± 50.3	14.67 ± 0.78	0.02
Amx + P7	343.23 ± 13.9	12.3 ± 1.2	0.03
Amx + P8	266.04 ± 23.06	24.32 ± 4.45	0.09
Amx + P9	353.34 ± 5.45	29.87 ± 5.7	0.06
Amx + P10	344.18 ± 29.09	16.03 ± 1.03	0.04
**SHV-14**	***K*_m_ (μM)**	***k_cat_* (s^−1^)**	***k_cat_*/*K*_m_ (μM)**
Pip	98.74 ± 12.16	432.09 ± 5.97	4.4
Pip + P5	373.98 ± 19.94	35.02 ± 7.01	0.09
Pip + P6	537.05 ± 8.97	31.21 ± 6.32	0.06
Pip + P7	424.01 ± 6.89	32.01± 6.43	0.03
Pip + P8	542.34 ± 11.32	34.13 ± 3.12	0.05
Pip + P9	332.28 ± 32.9	28.81 ± 0.1	0.06
Pip + P10	411.32 ± 15.01	18.24 ± 2.1	0.03

Footnotes: Pen: Penicillin, Amp: Ampicillin, Amx: Amoxicillin, Pip: Piperacillin.

## Data Availability

Data sharing not applicable.

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
