# Peer review of "Antimicrobial Peptides Designed against the Ω-Loop of Class A β-Lactamases to Potentiate the Efficacy of β-Lactam Antibiotics"

_antibiotics, 2023, doi:10.3390/antibiotics12030553_

Round 1
Reviewer 1 Report
The manuscript by Biswal et al. reports the design, synthesis and biological evaluation of antimicrobial peptides targeting class A beta-lactamases in attempt to reduce the beta-lactam resistance and enhance the efficacy of beta-lactam antibiotics. In this work, the authors considered the role of omega-loop in beta-lactam resistance and designed a group of peptides with the aid of in silico docking studies. Ten targeted peptides were then synthesised by solid phase peptide synthesis. Biological testing of these peptides suggested that peptide 5 and peptide 6 were the most potent peptides which enhanced the efficacy of beta-lactam antibiotics most. Mechanistic studies suggested that these peptides act via inhibiting beta-lactamases and this observation was supported by in silico molecular dynamic simulations.
Overall, this manuscript is informative and should be useful for others in the field. However, the following aspects have to be addressed before it is ready for publication:
1. The ten peptides synthesised and tested are referred as P1-P10 in the manuscript. The sequence of each peptide is only included in the non-published material file. The authors should tabulate the sequence of each peptide (P1-P10) in the manuscript.
2. Table 2 displays the same information as Table 3a and does not provide the docking results of the peptides.
3. In lines 12-14 on page 9, the authors claimed that “E. coli CS109 strains displayed substantially lower MICs for all the penicillin group of antibiotics tested in the presence of peptides than that of the control”. However, this is arguable. In the presence of P3, P5 and P8, penicillin showed the same MIC as the control. Similar cases are also observed for ampicillin, amoxicillin and piperacillin. In the presence of P3, ampicillin even showed a higher MIC value than the control, suggesting that this combination has a lower antibacterial activity than the control.
4. In line 18 on page 9, “P10: ~2-9 fold” should be “P10: ~2-8 fold”.
5. In lines 38-40, the authors mentioned that membrane disruption could have contributed to the antimicrobial activity of the peptides. I wonder if the authors have verified this (by cytoplasmic membrane permeability assay or otherwise)?
6. Table 3c is not mentioned until page 11. I would recommend the authors to move Table 3c to after line 61 on page 11.
7. In line 51 of page 11, for P5, it should be “~2-4 fold” instead of “~2-8 fold”. Similar mistakes are found elsewhere in the manuscript. The authors should check the manuscript carefully to fix all these mistakes.
8. Regarding Table S1,
a) I recommend moving it from the supplementary information file to the manuscript.
b) I wonder why P9 and P10 were not tested.
9. The plots in Figure 3 are too small and the resolution is too low to be read.
10. I wonder if the authors have assessed the cytotoxicity of the synthesised peptides against human cells.
11. The authors should insert the sub-heading “Conclusion” before the last paragraph on page 15.
12. All data in the non-published material file should be moved to the supplementary information file and to be published with the manuscript, as these data provide the purity of the synthesised peptides.
13. Peptide 2 and peptide 3 only have purity of 82%. The authors should be aware that all biologically tested compounds should have a purity of at least 95%.
Author Response
1*Comment: The ten peptides synthesised and tested are referred as P1-P10 in the manuscript. The sequence of each peptide is only included in the non-published material file. The authors should tabulate the sequence of each peptide (P1-P10) in the manuscript.
Response: This information has been included in the manuscript (Line 207-210).
2*Comment: Table 2 displays the same information as Table 3a and does not provide the docking results of the peptides.
Response: The tables have been corrected in the updated manuscript.
3*Comment: In lines 12-14 on page 9, the authors claimed that “E. coli CS109 strains displayed substantially lower MICs for all the penicillin group of antibiotics tested in the presence of peptides than that of the control”. However, this is arguable. In the presence of P3, P5 and P8, penicillin showed the same MIC as the control. Similar cases are also observed for ampicillin, amoxicillin and piperacillin. In the presence of P3, ampicillin even showed a higher MIC value than the control, suggesting that this combination has a lower antibacterial activity than the control.
Response: Although we concur with the reviewer, we wish to provide a thorough explanation of the table. Out of ten peptides, our main goal was to select the ones that demonstrated best antibacterial activity in combination with beta-lactam antibiotics.
In the presence of P3, P5 and P8, only penicillin showed the same MIC as the control:
P3: penicillin: no fold, ampicillin: no fold, amoxicillin: no fold, piperacillin: 2 fold difference
P5: penicillin: no fold, ampicillin: 2 fold, amoxicillin: 4 fold, piperacillin: 8 fold
P8: penicillin: no fold, ampicillin: no fold, amoxicillin: 2 fold, piperacillin: 4 fold
Here in the case of P3, there is only 2 fold difference i.e., only in the case of piperacillin, whereas in the case of P5 and P8 there is a fold difference observed in the case of two and three antibiotics. As a result, the P3 is eliminated, and P5, P8, were chosen for additional research.
4*Comment: In line 18 on page 9, “P10: ~2-9 fold” should be “P10: ~2-8 fold”.
Response: This correction has been incorporated in the manuscript (lines # 223-224)
5*Comment: In lines 38-40, the authors mentioned that membrane disruption could have contributed to the antimicrobial activity of the peptides. I wonder if the authors have verified this (by cytoplasmic membrane permeability assay or otherwise)?
Response: The authors agree with the reviewer on this. As we previously indicated, the molecular mechanism of action of these synthetic peptides has not been clarified in the current form of the publication because it requires a distinct set of investigations. We have concentrated on finding peptides that can suppress the action of class A beta-lactamases.
Performing the localization assay has certain issues. i.e., In order to perform the confocal/fluorescence microscopy we have to tag the protein with the reporter gene, which is a long task to do and it will take a long time. The GFP/RFP tag construction must first be made before testing. The third point is that even if the fluorescence-labelled peptides reach the periplasm, this does not necessarily indicate that they will bind to the beta-lactamases there. But if it binds, it might produce higher fluorescence; otherwise, it might diffuse.
6*Comment: Table 3c is not mentioned until page 11. I would recommend the authors to move Table 3c to after line 61 on page 11.
Response: Table 3c represents the MICs of beta-lactam drugs in presence of peptides in M. smegmatis. Line 61 on page 11 (according to mdpi format) contains the results obtained from M. tuberculosis. Therefore, if Table 3c is shifted to Line 61 on page 11 it will refer to results obtained from M. tuberculosis (which is incorrect) not M. smegmatis and the outcome of the session will change if Table 3c is moved to line 61 on page 11.
7*Comment: In line 51 of page 11, for P5, it should be “~2-4 fold” instead of “~2-8 fold”. Similar mistakes are found elsewhere in the manuscript. The authors should check the manuscript carefully to fix all these mistakes.
Response: These corrections have been incorporated into the manuscript (Line 242).
8*Comment: Regarding Table S1,
- a) I recommend moving it from the supplementary information file to the manuscript.
- b) I wonder why P9 and P10 were not tested.
Response: Regrettably due to limited time (due to covid restrictions) and the availability of proper resources we could not test P9 and P10. As a result, we have chosen to keep Table S1 as the supplemental material.
9*Comment: The plots in Figure 3 are too small and the resolution is too low to be read.
Response: We have revised the figure in the manuscript and have included it with a higher-resolution image.
10*Comment: I wonder if the authors have assessed the cytotoxicity of the synthesised peptides against human cells.
Response: We have not assessed the cytotoxicity of peptides in this proof-of-principle study as our focus was on determining the efficacy of the peptide/beta-lactam combinations in vitro. Cytotoxicity determination will be an important part of our future studies with these peptides.
11*Comment: The authors should insert the sub-heading “Conclusion” before the last paragraph on page 15.
Response: We have made this change in the revised manuscript.
12*Comment: All data in the non-published material file should be moved to the supplementary information file and to be published with the manuscript, as these data provide the purity of the synthesised peptides.
Response: We have included the peptide characterisation data with the main supplementary information file in the revised manuscript.
13*Comment: Peptide 2 and peptide 3 only have a purity of 82%. The authors should be aware that all biologically tested compounds should have a purity of at least 95%.
Response: The authors thank the reviewer and acknowledge the importance that tested compounds have high purity. We also acknowledge that the Arg-containing peptides 1-4 have lower purity, which may have contributed to the minimal decrease in MIC value we observed for these peptides. However, as stated in our study, the six peptides we prioritised for further testing (kinetics studies and MIC testing in E. coli overexpressing β-lactamases) were the histidine-containing peptides 5-10, all of which were obtained in moderate yield with purity >95%. Therefore, we believe that the conclusions drawn from the corresponding results are valid.
References cited while responding to the comments:
- Ma, B.; Fang, C.; Lu, L.; Wang, M.; Xue, X.; Zhou, Y.; Li, M.; Hu, Y.; Luo, X.; Hou, Z. The Antimicrobial Peptide Thanatin Disrupts the Bacterial Outer Membrane and Inactivates the NDM-1 Metallo-β-Lactamase. Commun. 2019, 10, 3517.
- Costa, B.O.; Cardoso, M.H.; Franco, O.L. Development of Peptides That Inhibit Aminoglycoside-Modifying Enzymes and β-Lactamases for Control of Resistant Bacteria. Protein Pept. Sci. 2020, 21, 1011–1026.
- Hou, Z.; Lu, J.; Fang, C.; Zhou, Y.; Bai, H.; Zhang, X.; Xue, X.; Chen, Y.; Luo, X. Underlying Mechanism of in Vivo and in Vitro Activity of C-Terminal--Amidated Thanatin against Clinical Isolates of Extended-Spectrum β-Lactamase--Producing Escherichia Coli. Infect. Dis. 2011, 203, 273–282.
- Hu, Y.; Liu, A.; Vaudrey, J.; Vaiciunaite, B.; Moigboi, C.; McTavish, S.M.; Kearns, A.; Coates, A. Combinations of β-Lactam or Aminoglycoside Antibiotics with Plectasin Are Synergistic against Methicillin-Sensitive and Methicillin-Resistant Staphylococcus Aureus. PLoS One 2015, 10, e0117664.
- Rishi, P.; Preet Singh, A.; Garg, N.; Rishi, M. Evaluation of Nisin--β-Lactam Antibiotics against Clinical Strains of Salmonella Enterica Serovar Typhi. Antibiot. (Tokyo). 2014, 67, 807–811.
- Dutta, M., Kar, D., Bansal, A., Chakraborty, S., & Ghosh, A. S. (2015). A single amino acid substitution in the Ω-like loop of E. Coli PBP5 disrupts its ability to maintain cell shape and intrinsic β-lactam resistance. Microbiology (United Kingdom), 161(4), 895–902.
- Höltje, J.-V. (1998). Growth of the stress-bearing and shape-maintaining murein sacculus of Escherichia coli. Microbiology and Molecular Biology Reviews, 62(1), 181–203
- Ghuysen, J. M. (1991). Serine β-lactamases and penicillin-binding proteins. Annual review of microbiology, 45(1), 37-67.
Reviewer 2 Report
Authors in this paper have attempted in identifying novel cationic peptides that can inhibit beta-lactamases and potentiate the activity of beta-lactam antibiotics. The work can be viewed as a substitute to the small-molecule beta-lactamase inhibitors and can be of significant potential. However, I have some basic concerns associated with this study and they are highlighted below-
Comments:
1. Referencing figure 1 in line 47, would be helpful.
2. Page 2, Line 59; where authors discuss examples of several peptides in the market. I personally feel it seems irrelevant to the context and throws of the readers in a different direction. It can be summarized in one sentence with different references. Also, authors should discuss alternative strategies employed to inhibit beta-lactamases, till date and their shortcomings, if any. This will help in elevate the importance of their objective, which I strongly believe is certainly interesting way to tackle beta-lactam resistance.
3. Authors should include few sentences in the result section on the design strategy of pentapeptides. It is not clear, what criteria was employed to decide on a pentapeptide library and not a hexapeptide or a heptapeptide, or any other template? Also, what considerations were made on deciding the amino acid composition of the library, so that it can function as a broad-spectrum beta-lactamase inhibitor? Did the authors cyclize the peptide for stability? Authors should include a table with the sequence of the designed peptides and their physicochemical properties ( net charge, pI, molecular weight).
4. Under the antimicrobial efficacy results section, authors should discuss about the type of beta-lactamases present in different strains used for the study. This will help in providing a context to the readers and understanding the broad nature of the peptide design.
5. Page 9, Line 5 to line 7 seems totally irrelevant to the context and should be eliminated.
6. In their MIC study, authors should include a control well in their assay, containing only peptide + bacteria combination, without the beta-lactam antibiotics. This will help in understanding if the peptides have any off-target action and demonstrate any antimicrobial property on its own.
7. In connection to the previous comment, authors have used a very low concentration of the peptide, which can be one of the reasons that they observe either none or only 2-4 fold change (only for few cases, ~2-8 fold) in the MIC of antibiotics in presence of the peptides. My recommendation would be to perform peptide titration to assess if they can achieve greater fold change with higher peptide concentration. In such case, taking into consideration my comment no.6 would be useful. Lastly, it is important to note that, for MIC experiments, a 2-fold change is not considered significant.
8. In table 2, authors should include a footnote on expanding the antibiotic abbreviations.
9. Authors discuss that the beta-lactamases localization are majorly in the periplasmic space. However, most of the cationic antimicrobial peptides localize either on the membrane creating pores or within the cytoplasm. So, it would be useful if authors investigate their peptide localization. If there is not enough peptide localization in the periplasmic space, it can be one of the other reasons to consider for the low fold-change in the MIC activity of beta-lactams.
10. It is a general observation that laboratory strains of bacteria are more susceptible to antibiotics/peptides in comparison to MDR strains. Authors in this paper reported similar observations with H37Rv laboratory strain. However, it is important to take into consideration that MDR strains adapt different resistant strategies, one of them being undergoing different membrane charge alterations to prevent the drug interaction and their diffusion through the membrane. Similar phenomenon can be taken into consideration here, which might be preventing peptide entry and can be cause of the lower fold change values. One way to assess is to evaluate the net membrane charge of the laboratory strain and some of the MDR strains.
11. Page 11, Line 82- ‘Hale and Hanlock revealed that cationic peptides could be mem- brane non-disruptive and potentially have many non-membrane targets owing to their amphiphilic nature (e.g., indolicidin)’.
This comment is true for certain antimicrobial peptides, which lack a structured conformation; otherwise, the general observation is that cationic AMPs are largely membrane-disruptive. Indolicidin is such an exception. So, re-clarifying this sentence would be better to make it more scientifically appropriate.
12. Under the results section ‘Cells expressing the β-lactamases confirmed enhanced sensitivity with peptides’, it is unclear why the authors overexpressed TEM-1 and SHV-14. Didn’t E. coli CS-109 already have endogenous beta-lactamase present in them? Some clarification is needed, why the earlier antimicrobial efficacy assay was performed and what did they try achieving with overexpression of additional beta-lactamases.
Also, the conclusions drawn from the results under this section is inappropriate. If the authors were interested in determining selectivity, the enzyme hydrolysis assay with Km and Kcat values for different classes of beta-lactamases in presence of peptides and the beta-lactam substrate is rather more convincing.
13. Page 13, Line 128 – Reference is missing.
14. Page 13, Line 139 – Reference to the data is missing.
15. Given my understanding that authors generated linear peptides, it will be useful to include a cyclized version of the six peptides and evaluate the MIC assay again to assess if the activity is enhanced further. Cyclization through disulfide bridges would provide better conformational rigidity and improved upon proteolytic stability.
16. Technical question: was the MIC and HT-SPOTi assay data reported from biological triplicate experiments? This piece of information is missing from the materials and methods section.
As a general comment, many of the conclusions drawn from different results is over-the-top and not well-supported by experimental data. Authors should tone down the discussion section a bit, where the conclusion are well-supported.
Author Response
1*Comment: Referencing figure 1 in line 47, would be helpful
Response: As per the suggestion, we have incorporated the reference of figure 1 (line 71).
2*Comment: Page 2, Line 59; where authors discuss examples of several peptides in the market. I personally feel it seems irrelevant to the context and throws of the readers in a different direction. It can be summarized in one sentence with different references. Also, authors should discuss alternative strategies employed to inhibit beta-lactamases, till date and their shortcomings, if any. This will help in elevate the importance of their objective, which I strongly believe is certainly interesting way to tackle beta-lactam resistance.
Response: We have taken the advice into account and written a single sentence to wrap up the previous information. However, the line presents some examples of short peptides relevant to the topic. In addition, we have supplied a few illustrations of peptide and small molecule inhibitors that are effective against beta-lactamases. The changes are added in the revised manuscript (line numbers 81-90) with references 12-16.
3*Comment: Authors should include a few sentences in the result section on the design strategy of pentapeptides. It is not clear, what criteria was employed to decide on a pentapeptide library and not a hexapeptide or a heptapeptide, or any other template? Also, what considerations were made on deciding the amino acid composition of the library, so that it can function as a broad-spectrum beta-lactamase inhibitor? Did the authors cyclize the peptide for stability? Authors should include a table with the sequence of the designed peptides and their physicochemical properties (net charge, pI, molecular weight).
Response: We acknowledge this point from the reviewer. The pentapeptides were linear and were designed based mostly on charge complementarity, allowing them to bind to the amino acids of the omega loop with potent electrostatic interactions. Additionally, factors like peptide length, solubility (the influence of hydrophobic and hydrophilic amino acids on solubility), hydropathy (charged/uncharged residues), and intermolecular H-bond forming capacity were analysed in detail before considering individual amino acids for the peptide sequences. This information has been added to the revised manuscript (line no:123-129). In addition, the supplementary material now includes a table (Table S1) which lists the sequences and various characteristics of peptides.
Several factors influenced our decision to design pentapeptides, including:
1: It has been seen that penicillin-binding proteins (PBPs) play a major role in peptidoglycan biosynthesis and have a great affinity for pentapeptides (Ghuysen, 1991; Holtje, 1998; Dutta et al., 2015). The PBPs, especially DD-carboxypeptidases, share a high degree of sequence similarity with beta-lactamases as they have a common ancestor. Therefore, the beta-lactamase may have an affinity towards pentapeptides as well.
2: Peptides with less than five residues are typically soluble in aqueous solutions, according to research (Peptide Design | Thermo Fisher Scientific - IN).
4*Comment: Under the antimicrobial efficacy results section, authors should discuss about the type of beta-lactamases present in different strains used for the study. This will help in providing a context to the readers and understanding the broad nature of the peptide design.
Response: In line with the Reviewer’s comment, we have incorporated some text in the revised manuscript (lines 287-290 and 293-294).
5*Comment: Page 9, Line 5 to line 7 seems totally irrelevant to the context and should be eliminated.
Response: The suggested lines were eliminated from the manuscript. Thank you.
6*Comment: In their MIC study, authors should include a control well in their assay, containing only peptide + bacteria combination, without the beta-lactam antibiotics. This will help in understanding if the peptides have any off-target action and demonstrate any antimicrobial property on its own.
Response: Thank you for your valuable feedback. We have incorporated the suggestions in the revised manuscript (lines no 259-263). We have also included a table in the supplemental material that exclusively lists the peptide + bacterium combinations (Table S3).
7*Comment: In connection to the previous comment, authors have used a very low concentration of the peptide, which can be one of the reasons that they observe either none or only 2-4, fold change (only for a few cases, ~2-8 fold) in the MIC of antibiotics in presence of the peptides. My recommendation would be to perform peptide titration to assess if they can achieve greater fold change with higher peptide concentration. In such a case, taking into consideration, my comment no.6 would be useful. Lastly, it is important to note that, for MIC experiments, a 2-fold change is not considered significant.
Response: In the current proof-of-principle study, we sought to ascertain only whether the designed peptides could potentiate β-lactam antibiotics by the designed mechanism when used in combination, and therefore selected to test the peptides at a single concentration and alongside β-lactam antibiotics. While we appreciate that potentiation and antimicrobial activity of the peptides could be concentration dependent and we must consider this in our continued investigation, we note that these experiments are beyond the scope of the current proof-of-principle study. We have added this interpretation as a statement in the discussion section.
8*Comment: In table 2, authors should include a footnote on expanding the antibiotic abbreviations.
Response: Thank you for your valuable feedback. In fact, Table 2 in the submitted word file describes the docking outcomes of peptides together with their binding energy. But it seems like the way in mdpi file format has been changed. The abbreviations of antibiotics are now spelt out in the footnote in the revised manuscript.
9*Comment: Authors discuss that the beta-lactamases localization is majorly in the periplasmic space. However, most of the cationic antimicrobial peptides localize either on the membrane creating pores or within the cytoplasm. So, it would be useful if authors investigate their peptide localization. If there is not enough peptide localization in the periplasmic space, it can be one of the other reasons to consider for the low fold-change in the MIC activity of beta-lactams.
Response: The cationic peptides inhibit the bacteria carrying beta-lactamases by disrupting the cell membrane as proven in the case of thanatin (Ma et al., 2019). However, in the present form of the manuscript, we have not included the elucidation of the molecular mechanism of action of these synthetic peptides, which will be followed by our further investigative study design and communicated in due course. We have concentrated on finding peptides that can suppress the action of class A beta-lactamases. We do agree with the reviewer that the localization of peptides can be one of the reasons for the low fold change in MICs of beta-lactams since we have observed a significant decrease in the in vitro beta-lactamase activity of enzymes in presence of peptides.
10*Comment: It is a general observation that laboratory strains of bacteria are more susceptible to antibiotics/peptides in comparison to MDR strains. Authors in this paper reported similar observations with H37Rv laboratory strain. However, it is important to take into consideration that MDR strains adapt different resistant strategies, one of them being undergoing different membrane charge alterations to prevent the drug interaction and their diffusion through the membrane. Similar phenomenon can be taken into consideration here, which might be preventing peptide entry and can be cause of the lower fold change values. One way to assess is to evaluate the net membrane charge of the laboratory strain and some of the MDR strains.
Response: We concur with the reviewer that laboratory strains of bacteria respond to antibiotics and peptides more readily than MDR strains (as they apply different drug-resistant strategies). However, in the present work, we sought to provide proof of principle that a group of peptides designed to bind the omega loop of beta-lactamases may decrease the enzyme's effectiveness and so amplify the impact of beta-lactam antibiotics. Though the idea given by the reviewer is very informative, the study of the molecular mechanism is beyond the scope of this study. In subsequent research, we will take this into consideration when deciding which of the six peptides will serve as the most effective peptide inhibitor.
11*Comment: Page 11, Line 82- ‘Hale and Hanlock revealed that cationic peptides could be mem- brane non-disruptive and potentially have many non-membrane targets owing to their amphiphilic nature (e.g., indolicidin)’. This comment is true for certain antimicrobial peptides, which lack a structured conformation; otherwise, the general observation is that cationic AMPs are largely membrane-disruptive. Indolicidin is such an exception. So, re-clarifying this sentence would be better to make it more scientifically appropriate.
Response: The correction has been incorporated in the manuscript (lines 284-285).
12*Comment: Under the results section ‘Cells expressing the β-lactamases confirmed enhanced sensitivity with peptides’, it is unclear why the authors overexpressed TEM-1 and SHV-14. Didn’t E. coli CS-109 already have endogenous beta-lactamase present in them? Some clarification is needed, why the earlier antimicrobial efficacy assay was performed and what did they try achieving with overexpression of additional beta-lactamases.
Also, the conclusions drawn from the results under this section is inappropriate. If the authors were interested in determining selectivity, the enzyme hydrolysis assay with Km and Kcat values for different classes of beta-lactamases in presence of peptides and the beta-lactam substrate is rather more convincing.
Response: In the present work, the peptides were designed to target the Class A beta-lactamases (TEM-1 and SHV-14) which exhibit resistance majorly to the penicillin group of beta-lactams. E. coli CS109 possess a chromosomal Class C beta-lactamase, AmpC which imparts intrinsic basal level resistance to some of the first-generation cephalosporins. In the present work, the beta-lactamases TEM-1 and SHV-14 were overexpressed in this host. The overexpression of these beta-lactamases overcomes the effect of the basal level resistance imparted by the AmpC since these beta-lactamases are more potent and over-expressed. Moreover, the antibiotics used in the present study include the penicillin group of beta-lactams, which is not included in the spectrum of activity of AmpC beta-lactamase of E. coli CS109.
To understand the impact of the peptides on the beta-lactamases under in vitro conditions, i.e., whether the peptides are acting in the same way as they have resulted under cellular studies, we have determined the kinetic parameters for β-lactam substrates.
We do not believe that the enzyme hydrolysis assay for different classes of beta-lactamases in presence of peptides and the beta-lactam substrate won’t be more convincing, because the peptides are designed specifically against the omega loop of class A beta-lactamases (not class B, class C and class D beta-lactamases). Furthermore, the omega-loop sequence and structure of distinct beta-lactamase classes differ. Therefore, the peptides designed against class A may not be effective against class B, class C and class D beta-lactamases.
13*Comment: Page 13, Line 128 – Reference is missing.
Response: The correction has been incorporated in the manuscript (Line 318).
14*Comment: Page 13, Line 139 – Reference to the data is missing.
Response: The correction has been incorporated in the manuscript (lines 327-329).
15*Comment: Given my understanding that authors generated linear peptides, it will be useful to include a cyclized version of the six peptides and evaluate the MIC assay again to assess if the activity is enhanced further. Cyclization through disulfide bridges would provide better conformational rigidity and improved upon proteolytic stability.
Response: The authors agree with the reviewer that cyclisation frequently improves peptide stability and often also binding affinity. However, the current study aimed only to demonstrate initial proof-of-principle that peptides binding the omega loop of β-lactamases can reduce the efficacy of the enzyme and therefore potentiate the effect of β-lactam antibiotics. As such, further optimisation of the presented peptides by cyclisation is beyond the scope of the current study. The authors thank the reviewer for this suggestion, however, and future studies on optimisation of the peptides will include cyclisation by disulfide bridging. Crosslinks such as thioethers which are more metabolically stable will also be considered.
16*Comment: Technical question: was the MIC and HT-SPOTi assay data reported from biological triplicate experiments? This piece of information is missing from the materials and methods section.
Response: All the experiments were conducted in triplicate and the correction has been incorporated into the manuscript (line number 168-169,178).
References cited while responding to the comments:
- Ma, B.; Fang, C.; Lu, L.; Wang, M.; Xue, X.; Zhou, Y.; Li, M.; Hu, Y.; Luo, X.; Hou, Z. The Antimicrobial Peptide Thanatin Disrupts the Bacterial Outer Membrane and Inactivates the NDM-1 Metallo-β-Lactamase. Commun. 2019, 10, 3517.
- Costa, B.O.; Cardoso, M.H.; Franco, O.L. Development of Peptides That Inhibit Aminoglycoside-Modifying Enzymes and β-Lactamases for Control of Resistant Bacteria. Protein Pept. Sci. 2020, 21, 1011–1026.
- Hou, Z.; Lu, J.; Fang, C.; Zhou, Y.; Bai, H.; Zhang, X.; Xue, X.; Chen, Y.; Luo, X. Underlying Mechanism of in Vivo and in Vitro Activity of C-Terminal--Amidated Thanatin against Clinical Isolates of Extended-Spectrum β-Lactamase--Producing Escherichia Coli. Infect. Dis. 2011, 203, 273–282.
- Hu, Y.; Liu, A.; Vaudrey, J.; Vaiciunaite, B.; Moigboi, C.; McTavish, S.M.; Kearns, A.; Coates, A. Combinations of β-Lactam or Aminoglycoside Antibiotics with Plectasin Are Synergistic against Methicillin-Sensitive and Methicillin-Resistant Staphylococcus Aureus. PLoS One 2015, 10, e0117664.
- Rishi, P.; Preet Singh, A.; Garg, N.; Rishi, M. Evaluation of Nisin--β-Lactam Antibiotics against Clinical Strains of Salmonella Enterica Serovar Typhi. Antibiot. (Tokyo). 2014, 67, 807–811.
- Dutta, M., Kar, D., Bansal, A., Chakraborty, S., & Ghosh, A. S. (2015). A single amino acid substitution in the Ω-like loop of E. Coli PBP5 disrupts its ability to maintain cell shape and intrinsic β-lactam resistance. Microbiology (United Kingdom), 161(4), 895–902.
- Höltje, J.-V. (1998). Growth of the stress-bearing and shape-maintaining murein sacculus of Escherichia coli. Microbiology and Molecular Biology Reviews, 62(1), 181–203
- Ghuysen, J. M. (1991). Serine β-lactamases and penicillin-binding proteins. Annual review of microbiology, 45(1), 37-67.
Reviewer 3 Report
The manuscript entitled „Antimicrobial peptides designed against the ῼ loop of class A 2 β-lactamases to potentiate the efficacy of β-lactam antibiotics” by Biswal et al. describes the enhancing antibacterial effect of various peptides in combination with antibiotics.
The study is interesting and materials and methods described in detail.
However, there are several weaknesses in this study, in particular relating to the presentation of the data. This makes it difficult to follow the study during reading and to recognize the impact of the findings.
Major points:
· Table 3 is accidently shown twice. Table 2 shows the antimicrobial activity but should show the docking results instead. Also, the manuscript lacks the sequences of the 10 peptides. The sequences should be added to Figure 2. Therefore, the sequences of peptides P1-P10 were only available for this review from the non-published word-file.
· Figure 1 is much too large, followed by one empty page
· The manuscript shows several series of tables and it is difficult to extract the results from them. It may be beneficial, if some of the tables are shown in the supplementary information and relevant findings should be rather represented by bar graphs or others. This would make it much easier for the reader to follow.
· The authors say that the effect of peptides with antibiotic is synergistic (line 112-113). What is the basis for that, was there any calculation of synergy performed?
· Finally, a conclusion is missing – which peptides should be further pursued and why? What are the next steps?
I would recommend that the authors should carefully revise their manuscript before submission.
Author Response
1*Comment: Table 3 is accidentally shown twice. Table 2 shows the antimicrobial activity but should show the docking results instead. Also, the manuscript lacks the sequences of the 10 peptides. The sequences should be added to Figure 2. Therefore, the sequences of peptides P1-P10 were only available for this review from the non-published word file.
Response: The tables have been corrected and the sequences of the peptides have been included in the text of the manuscript (Lines 208-210).
2*Comment: Figure 1 is much too large, followed by one empty page
Response: The picture size is decreased as suggested.
3*Comment: The manuscript shows several series of tables and it is difficult to extract the results from them. It may be beneficial, if some of the tables are shown in the supplementary information and relevant findings should be rather represented by bar graphs or others. This would make it much easier for the reader to follow.
Response: We have added two tables in the supplementary material (Table S1 and Table S3). Thank you.
4*Comment: The authors say that the effect of peptides with antibiotics is synergistic (lines 112-113). What is the basis for that, was there any calculation of synergy performed?
Response: We have used the word relating to the combinatory effect of both peptides and beta-lactam drugs as they both produced some coactive effect. The synergy calculation was not performed. As per the suggestion, instead of resistance, we have incorporated the terms effective (Line 311) and combinational (line 48) in the manuscript.
5*Comment: Finally, a conclusion is missing – which peptides should be further pursued and why? What are the next steps?
Response: We have incorporated a section as a subheading “conclusion” in the revised manuscript (Line number 358, 361-362, 364, 367-373).
From the study, out of 10 peptides, six were found to display more bactericidal activities. Therefore, in our next phase of study we will focus on these 6 peptides to find the most promising one. Further studies including the cyclization of peptides, cytotoxicity assay and peptide localization study will be required to identify the best inhibiting peptide among the six shortlisted peptides.
References cited while responding to the comments:
- Ma, B.; Fang, C.; Lu, L.; Wang, M.; Xue, X.; Zhou, Y.; Li, M.; Hu, Y.; Luo, X.; Hou, Z. The Antimicrobial Peptide Thanatin Disrupts the Bacterial Outer Membrane and Inactivates the NDM-1 Metallo-β-Lactamase. Commun. 2019, 10, 3517.
- Costa, B.O.; Cardoso, M.H.; Franco, O.L. Development of Peptides That Inhibit Aminoglycoside-Modifying Enzymes and β-Lactamases for Control of Resistant Bacteria. Protein Pept. Sci. 2020, 21, 1011–1026.
- Hou, Z.; Lu, J.; Fang, C.; Zhou, Y.; Bai, H.; Zhang, X.; Xue, X.; Chen, Y.; Luo, X. Underlying Mechanism of in Vivo and in Vitro Activity of C-Terminal--Amidated Thanatin against Clinical Isolates of Extended-Spectrum β-Lactamase--Producing Escherichia Coli. Infect. Dis. 2011, 203, 273–282.
- Hu, Y.; Liu, A.; Vaudrey, J.; Vaiciunaite, B.; Moigboi, C.; McTavish, S.M.; Kearns, A.; Coates, A. Combinations of β-Lactam or Aminoglycoside Antibiotics with Plectasin Are Synergistic against Methicillin-Sensitive and Methicillin-Resistant Staphylococcus Aureus. PLoS One 2015, 10, e0117664.
- Rishi, P.; Preet Singh, A.; Garg, N.; Rishi, M. Evaluation of Nisin--β-Lactam Antibiotics against Clinical Strains of Salmonella Enterica Serovar Typhi. Antibiot. (Tokyo). 2014, 67, 807–811.
- Dutta, M., Kar, D., Bansal, A., Chakraborty, S., & Ghosh, A. S. (2015). A single amino acid substitution in the Ω-like loop of E. Coli PBP5 disrupts its ability to maintain cell shape and intrinsic β-lactam resistance. Microbiology (United Kingdom), 161(4), 895–902.
- Höltje, J.-V. (1998). Growth of the stress-bearing and shape-maintaining murein sacculus of Escherichia coli. Microbiology and Molecular Biology Reviews, 62(1), 181–203
- Ghuysen, J. M. (1991). Serine β-lactamases and penicillin-binding proteins. Annual review of microbiology, 45(1), 37-67.
Reviewer 4 Report
The manuscript „ Antimicrobial peptides designed against the ῼ loop of class A β-lactamases to potentiate the efficacy of β-lactam antibiotics” is a well-written study summarizing the efforts into the development of peptides expected to act as novel inhibitors that can be used specifically against the Ω-loop of the β-lactamases. The research concept is well-considered, and the methodology is accurately selected to prove the hypothesis. What I would suggest to improve is as follows:
- The Authors should pay attention to provide a full name when they introduce an abbreviation for the first time (e.g. ESBL).
- I would suggest separating the last paragraph of R&D section as a „Conclusion” section.
- The numerical data presenting the sensitivity of cells to the β-lactam drugs (page 9, lines 12-22) should be double-checked.
- The Authors could consider presenting some of the numerical data on graphs, e.g. content of Table 6.
- The names of axis and nubers presented in Figure 3 are not visible, therefore this figure should be revised to make sure that all the information can be easily tracked.
Author Response
1*Comment: The Authors should pay attention to provide a full name when they introduce an abbreviation for the first time (e.g., ESBL).
Response: In line with the suggestion, we have incorporated changes to the manuscript (Line 94).
2*Comment: I would suggest separating the last paragraph of R&D section as a „Conclusion” section.
Response: The last paragraph was separated and in the updated manuscript, we labelled it "conclusion." (Line number 358).
3*Comment: The numerical data presenting the sensitivity of cells to the β-lactam drugs (page 9, lines 12-22) should be double-checked.
Response: Suggestions have been incorporated into the manuscript.
4*Comment: The Authors could consider presenting some of the numerical data on graphs, e.g., content of Table 6.
Response: We agree with the reviewer that the graphical representation of the data depicts the importance of the values. However, the tabular representation of the kinetic data is required for the elucidation of all three parameters such as kcat, Km, kcat/Km simultaneously, which could not be expressed in a comprehensive way using the graphical representation. The graphical presentation of the data makes it more challenging to provide a proper explanation for the enzyme affinity (how many times the Km is increasing/decreasing in the presence of peptides). Similarly, the Kcat values depicted graphically cannot be determined with any degree of accuracy.
5*Comment: The names of axis and numbers presented in Figure 3 are not visible, therefore this figure should be revised to make sure that all the information can be easily tracked.
Response: Figure 3 has been revised in the manuscript.
References cited while responding to the comments:
- Ma, B.; Fang, C.; Lu, L.; Wang, M.; Xue, X.; Zhou, Y.; Li, M.; Hu, Y.; Luo, X.; Hou, Z. The Antimicrobial Peptide Thanatin Disrupts the Bacterial Outer Membrane and Inactivates the NDM-1 Metallo-β-Lactamase. Commun. 2019, 10, 3517.
- Costa, B.O.; Cardoso, M.H.; Franco, O.L. Development of Peptides That Inhibit Aminoglycoside-Modifying Enzymes and β-Lactamases for Control of Resistant Bacteria. Protein Pept. Sci. 2020, 21, 1011–1026.
- Hou, Z.; Lu, J.; Fang, C.; Zhou, Y.; Bai, H.; Zhang, X.; Xue, X.; Chen, Y.; Luo, X. Underlying Mechanism of in Vivo and in Vitro Activity of C-Terminal--Amidated Thanatin against Clinical Isolates of Extended-Spectrum β-Lactamase--Producing Escherichia Coli. Infect. Dis. 2011, 203, 273–282.
- Hu, Y.; Liu, A.; Vaudrey, J.; Vaiciunaite, B.; Moigboi, C.; McTavish, S.M.; Kearns, A.; Coates, A. Combinations of β-Lactam or Aminoglycoside Antibiotics with Plectasin Are Synergistic against Methicillin-Sensitive and Methicillin-Resistant Staphylococcus Aureus. PLoS One 2015, 10, e0117664.
- Rishi, P.; Preet Singh, A.; Garg, N.; Rishi, M. Evaluation of Nisin--β-Lactam Antibiotics against Clinical Strains of Salmonella Enterica Serovar Typhi. Antibiot. (Tokyo). 2014, 67, 807–811.
- Dutta, M., Kar, D., Bansal, A., Chakraborty, S., & Ghosh, A. S. (2015). A single amino acid substitution in the Ω-like loop of E. Coli PBP5 disrupts its ability to maintain cell shape and intrinsic β-lactam resistance. Microbiology (United Kingdom), 161(4), 895–902.
- Höltje, J.-V. (1998). Growth of the stress-bearing and shape-maintaining murein sacculus of Escherichia coli. Microbiology and Molecular Biology Reviews, 62(1), 181–203
- Ghuysen, J. M. (1991). Serine β-lactamases and penicillin-binding proteins. Annual review of microbiology, 45(1), 37-67.
Round 2
Reviewer 1 Report
The authors have addressed most of the issues raised by the reviewers and amended the manuscript accordingly. However, the following aspects are yet to be resolved:
1. Table 2 remains unchanged and still not displaying the docking results of the peptides.
2. While the authors agreed on point #3 raised by reviewer #1, no changes have been made to the manuscript. The authors should rewrite the corresponding sentence as following:
“E. coli CS109 strains generally displayed lower MICs for all the penicillin group of antibiotics tested in the presence of peptides than that of the control”.
3. It is important that all biologically tested compounds should have a purity of 95%. The authors argued that the lower purity of peptides 1-4 may have contributed to the minimal decrease in MIC value. However, this argument is not true for all cases. The impurities might possess a significant lower or higher antibacterial activity against bacteria compared to the peptide. Unless the authors isolate the impurities in peptides 1-4 and test their activity against the bacteria, the authors cannot draw the above conclusion. If the authors cannot purify peptides 1-4 and test the purified peptides biologically, they should at least state in the manuscript that peptides 1-4 do NOT have purities of >95%, so that the readers can decide themselves whether the biological results are trustable.
Author Response
- Table 2 remains unchanged and still not displaying the docking results of the peptides.
Response: Revised Table 2 is incorporated directly into the re-revised manuscript (Page number 8, Lines #1-9).
- While the authors agreed on point #3 raised by reviewer #1, no changes have been made to the manuscript. The authors should rewrite the corresponding sentence as following:
“E. coli CS109 strains generally displayed lower MICs for all the penicillin group of antibiotics tested in the presence of peptides than that of the control”.
Response; Suggested sentence is now incorporated in the re-revised manuscript.
- It is important that all biologically tested compounds should have a purity of 95%. The authors argued that the lower purity of peptides 1-4 may have contributed to the minimal decrease in MIC value. However, this argument is not true for all cases. The impurities might possess a significant lower or higher antibacterial activity against bacteria compared to the peptide. Unless the authors isolate the impurities in peptides 1-4 and test their activity against the bacteria, the authors cannot draw the above conclusion. If the authors cannot purify peptides 1-4 and test the purified peptides biologically, they should at least state in the manuscript that peptides 1-4 do NOT have purities of >95%, so that the readers can decide themselves whether the biological results are trustable.
Response: A sentence is added on Page 6, lines #94-95: “However, we have only managed to achieve less than 95% purity for the peptides 1-4.”
Reviewer 2 Report
The authors have addressed all the concerns. I recommend "accept in present form".
Author Response
We thank the reviewer.
Reviewer 3 Report
The authors have improved their manuscript according to the reviewer´s suggestions. The peptide sequences have now been added to the text. However, one major point was the incorrect Table 2 which should show the docking results. This table has not been replaced in the uploaded version 2 of the manuscript. This has to be corrected.
Although I still feel that the data could be presented in a better way (e. g. by showing bar graphs instead of several tables), the manuscript can be considered for publication after correction of Table 2.
Author Response
Comments and Suggestions for Authors
The authors have improved their manuscript according to the reviewer´s suggestions. The peptide sequences have now been added to the text. However, one major point was the incorrect Table 2 which should show the docking results. This table has not been replaced in the uploaded version 2 of the manuscript. This has to be corrected.
Although I still feel that the data could be presented in a better way (e. g. by showing bar graphs instead of several tables), the manuscript can be considered for publication after the correction of Table 2.
Response: Revised Table 2 has been directly attached to the re-revised manuscript now (Page number 8, Lines #1-9).
Round 3
Reviewer 1 Report
The authors have addressed all issues raised by reviewers and I would recommend the publication of the manuscript.
Reviewer 3 Report
The authors have now corrected Table 2 in version 3 of the manuscript. The paper is now acceptable for publication.